

**Improving vegetation phenological parameterization of a land**
**surface model**
**Baozhang Chen[1,2,3] and Mingliang Che[3]**
[1]Key Laboratory of Soil and Water Conservation and Desertification Combating,
Ministry of Education,Beijing Forestry University, Beijing 100083,China
[2]China University of Mining and Technology, Xuzhou, and Jiangsu Center for
Collaborative Innovation in Geographic Information Resource Development and
Application, Nanjing, China
[3]State Key Laboratory of Resources and Environmental Information System, Institute
of Geographical Sciences and Natural Resources Research, Chinese Academy of
Sciences, Beijing
Correspondence to Baozhang Chen (Baozhang.Chen@igsnrr.ac.cn) and Mingliang
Che (chemingliangs@163.com)



**Abstract**: The growing degree day (GDD) model and the growing season
index (GSI) model are two common approaches used in various land surface
models (LSMs) for simulating phenophases. The capacity of these two
models for simulating phenolphases was evaluated by coupling them to a
LSM (DLM: Dynamic Land Model) and validated by observation data from
the 22 selected eddy covariance flux towers representing six typical plant
functional types. The main findings are threefold: (i) the simulated
phenophases using DLM-GSI were much closer to the observations derived
from the green chromatic coordinate data than using DLM-GDD. The start
of the growing season (SGS) was estimated to be earlier by DLM-GSI and
later by DLM-GDD. Meanwhile, the end of growing season (EGS) was
estimated to be later by DLM-GSI and earlier by DLM-GDD; (ii) compared
to the GDD model, the GSI model significantly decreased the absolute bias
of the phenophases simulated by DLM for all sites. The DLM-GSI model
simulated biases for SGS and EGS decreased by 48.2% and by 39% on
average, respectively; and (iii) the accuracy of modeled GPP using the
DLM-GSI model is much higher than using the DLM-GDD model for all
sites. The DLM-GSI model reduced the root mean square error of simulated
GPP by 8.0% and increased the corresponding index of agreement by 7.5%.





## 1 Introduction

Vegetation phenology is the timing of biological events in plants that is
influenced by environmental conditions, especially by long-term temperature changes
(Schwartz 2013). Phenology not only reflects the seasonal alternation but also the
adaptability of vegetation to environmental conditions (Che et al. 2014a). With a rapid
global climate change, the phenology of vegetation has adjusted to ensure survival and
reproduction (Eastman et al. 2013), and these changes have become the most sensitive
indicator of climate change (Cong et al. 2012; Hamunyela et al. 2013; Menzel; Fabian
1999; Schwartz 1998). Approaches for depicting phonological changes have been
recently employed in land surface models (LSMs) and have been coupled to global
circulation models for estimating the effects of climate change and accounting for
possible feedback (Subin et al. 2011). In LSMs, phenology is a very important module,
which controls the changes and length of the growing season and influences the
carbon cycle, evapotranspiration and the energy balance in the vegetation canopy
(Knorr et al. 2010; Kucharik; Twine 2007; White et al. 2009). Therefore, accurately
estimating phenophases in a LSM is critical to simulating the interactions between
terrestrial ecosystems and climate change.
Phenological approaches in LSMs can be divided into two categories. One is
satellite phenological observation, *i.e.*, the use of remotely sensed leaf area index
(LAI), which describes changes in the vegetation growing season and provides a



spatially integrative view of continuous biophysical states (Stöckli et al. 2008). For
instance, in the community land model (CLM) (Oleson et al. 2013; Oleson et al. 2010)
and the ecosystem-atmosphere simulation scheme (EASS) (Chen et al. 2007), the LAI
is read directly to characterize the effects of the three-dimensional canopy structure on
radiation, energy and carbon fluxes. Another is the process-based phenology model,
which is embedded in LSMs either explicitly, implicitly or both. Explicit phenology
models are independent of LSMs and are usually driven by offline climate factors.
The growing degree day (GDD) model and the growing season index (GSI) model are
the two common used representative explicit models.

The GDD model starts from Reaumur's approach, which first introduced the

concept of the degree-day sum and later became referred to as the thermal time model
(TM) or the spring warming model (WM) (Schwartz 2013). Chuine's approach
replaced the TM model because it introduced chilling requirements in dormancy and
unified various models that described the relationships between the temperature and
the rate of forcing and chilling development (Chuine 2000). In Chuine's approach, the
state of forcing was described as an accumulated number of the growing degree day
(Murray et al. 1989), and the state of chilling was also described as an accumulated
numbers of the chilling or freezing day (CD or FD). Applying the GDD and CD
approaches to the initiation of leaf onset has gained considerable recognition (Arora;
Boer 2005). However, this model is unable to simulate the reversible nature of the
spring recovery; cold temperatures during late spring can cause growing plants to



suffer from substantial cold damage (Arora; Boer 2005). In addition to temperature,
water-stress and photoperiod are also considered important factors in vegetation
phenology (Borchert et al. 2005). Studies of GDD models incorporating the effects of
soil water and photoperiod have been published (Caffarra et al. 2011; Lawrence et al.
2011). Currently, GDD models were employed by many LSMs, such as the version 4
of CLM (CLM4, CLawrence et al. 2011), the biome-BGC model (Thornton;
Rosenbloom 2005; Thornton et al. 2002), the integrated biosphere simulator (IBIS)
(Foley et al. 1996; Kucharik 2003), the lund-potsdam-jena model (LPJ) (Sitch et al.
2000; Sitch et al. 2003) and the IAP dynamic global vegetation model (IAP-DGVM)
(Zeng et al. 2014).

The GSI model combined a number of climate factors closely related to

phenology, *e.g.,* temperature, light and humidity, into an index to quantify the
greenness of vegetation throughout the year (Jolly et al. 2005). This approach is
simple and generalized to describe phenological states on local across global scales.
Furthermore, this approach is flexible enough to introduce other phenological
influence factors (PIFs), as long as the relationship between the PIF and plant growing
state can be reasonably described. The GSI model has not yet been employed by any
published LSMs. In this study, it was coupled to a LSM (DLM : the dynamic land
surface model ).    The DLM model was further developed by combining the
algorithms embedded in EASS and CLM4 to simulate biological, geographical,
physical and chemical processes (Chen et al. 2014; Chen et al. 2013) and was evolved



into a phenology module for simulating the seasonal changes in vegetation growth.
Most studies of phenological estimates focusing on the phenology (RSP)
retrieval algorithms based on remote sensing data, *e.g.,* LAI or normalized difference
vegetation index (NDVI) (White et al. 2014; White et al. 2009). However, published
studies that comparing process-based phenology models are limited, and researched
on evaluating the phenology models coupled into LSMs are even rarer. This gap
means that the validity of phenology simulation in LSMs is debatable and increases
uncertainty in the estimation of carbon, water and energy exchanges in LSMs.
In this study, we compared the performance of two common used phenology
models, GSI and GDD phenology models, which were coupled into DLM, focusing
on two vegetation types: deciduous forest and grass. The accuracy of the phenological
simulations in the two versions of DLM was evaluated against observations.
Moreover, another very important variable closely related to phenology, gross primary
production (GPP), was also simulated and analyzed.
**2 Methods and materials**
**2.1 Model descriptions**
**2.1.1 Outline of the DLM model**
The DLM model is prognostic and has been coupled to CESM 1.0.3 (Chen et al.
2014; Chen et al. 2013). DLM builds on EASS and CLM4. The main differences in
algorithms among DLM, EASS, and CLM4 are shown in Table 1. DLM absorbed the



vegetation physiological and physical algorithms based on the two-leaf canopy model,
which can effectively address radiation transfer through the canopy and its impact on
carbon sequestration and energy partitioning in EASS (Chen et al. 2007). DLM also
employs the plant and soil biochemical processes algorithms from CLM4, which
amply describe the relevant mechanisms, especially in the carbon-nitrogen (CN)
biogeochemical module.
**2.1.2 Phenological modules in DLM**
2.1.2.1 Growing season index module
The growing season index (GSI) model uses the GSI and corresponding criteria
for phenological transition stages to track all leaf phenological states and does not
need to distinguish the deciduous vegetation types.
(1) Growing season index
The growing season index for triggering the leaf green-up and defoliation (Jolly et
al. 2005; Stöckli et al. 2008) is expressed as:
$$GSI = f(T) \cdot f(DL) \cdot f(VPD) \qquad (1)$$

where $GSI$ has no unit, and its value varies from 0 to 1. The parameters $f(T)$, $f(DL)$ and
$f(VPD)$ are the temperature index, the day length index and the vapor pressure deficit
(VPD) index, respectively. They have no units and with values of 0~1. The statistics
shows that the GSI is positively correlated with the NDVI or LAI very significantly.
The temperature index $f(T)$ is calculated as,





$$f(T) = \begin{cases} 0, & T \leq T_{min} \\ \dfrac{T - T_{min}}{T_{max} - T_{min}}, & T_{min} < T < T_{max} \\ 1, & T \geq T_{max} \end{cases} \qquad (2)$$

where $T$, $T_{min}$ and $T_{max}$ are the temperature and the minimum and maximum
temperature thresholds in degrees K, respectively.

The day length index $f(DL)$ is calculated as,

$$f(DL) = \begin{cases} 0, & DL \leq DL_{min} \\ \dfrac{DL - DL_{min}}{DL_{max} - DL_{min}}, & DL_{min} < DL < DL_{max} \\ 1, & DL \geq DL_{max} \end{cases} \qquad (3)$$

where $DL$, $DL_{min}$ and $DL_{max}$ are the day length and the minimum and maximum of the
day length thresholds in hours, respectively.

The vapor pressure deficit index $f(VPD)$ is calculated as,

$$f(VPD) = \begin{cases} 0, & VPD \geq VPD_{max} \\ 1 - \dfrac{VPD - VPD_{min}}{VPD_{max} - VPD_{min}}, & VPD_{min} < VPD < VPD_{max} \\ 1, & VPD \leq VPD_{min} \end{cases} \qquad (4)$$

where $VPD$, $VPD_{min}$ and $VPD_{max}$ are the vapor pressure deficit and the minimum and
maximum VPD thresholds in $Pa$, respectively.

(2) Phenology strategy

There are four leaf phenology states in the GSI model: green-up (*i.e.*, the start of

the growing season, SGS, or start of season, SOS), normal growth, defoliation and
dormancy (*i.e.*, the end of the growing season, EGS, or end of season, EOS). Fig. 1
shows the corresponding method for extracting phenophases.





At the end of vegetation dormancy, when environmental conditions become

favorable to growth, the vegetation starts to emerge from dormancy and grow. To
trigger vegetation green-up, the *GSI* must be smoothed by a 21-day forward moving
average filter first. The moving average serves to buffer single extreme events from
prematurely triggering canopy changes (Jolly et al. 2005). Then, the accumulated GSI
approach for triggering the green-up is applied as follows:
$$GSIG_{sum} = \begin{cases} GSIG_{sum}^{pre} + f_{day}, & GSI \geq GSIG_{thr} \\ 0, & GSI < GSIG_{thr} \end{cases}$$
(5)

where $f_{day} = \mathrm{V}t / 86400$, $\mathrm{V}t$ is time step and is equal to 1800 sec. The superscript *pre*
represents the last time step. $GSIG_{sum}$ is the GSI summation for green-up in days, and
$GSIG_{thr}$ has no unit for the GSI threshold for green-up.

When $GSIG_{sum} > 6$, the leaf green-up begins, and the onset counter for

controlling the green-up length ($t_{onset}$, day) is initialized. Here, the criterion "$GSIG_{sum} >$
*6*" is followed by the leaf-out model in spring in Canadian terrestrial ecosystem model
(CTEM) (Arora; Boer 2005). In CTEM, the leaf-out state is triggered when the net
photosynthesis rate remains positive over 5-7 consecutive days. This criterion buffers
single extreme events from prematurely triggering canopy changes.

During the green-up period, the onset counter $t_{onset}$ is decremented at each time

step if $GSI \geq GSIG_{thr}$ until it reaches zero, then normal growth is triggered;
$$t_{onset} = \begin{cases} t_{onset}^{pre} - f_{day}, & GSI \geq GSIG_{thr} \\ t_{onset}^{pre}, & GSI < GSIG_{thr} \end{cases}$$
(6)

During normal growth, the vegetation grows stably, and its LAI gradually



reaches an annual peak. As adverse environmental conditions arrive in autumn,
vegetation enters the end of normal growth (*i.e.*, the start of defoliation) when
vegetation starts to drop leaves. To track leaf drop, an accumulated GSI approach is
used:
$$GSID_{sum} = \begin{cases} GSID_{sum}^{pre} + f_{day}, & GSI < GSID_{thr} \\ 0 & , & GSI \geq GSID_{thr} \end{cases} \qquad (7)$$

where $GSID_{sum}$ is the GSI summation for defoliation in days. The superscript *pre*
represents the last time step. $GSID_{thr}$ has no unit and is the GSI threshold for
defoliation.
When $GSID_{sum} > 6$, leaf defoliation is triggered, and the offset counter for
controlling the defoliation length ($t_{offset}$, day) is initialized. Here, the criterion
"$GSID_{sum} > 6$" uses the leaf-fall model in autumn in CTEM for reference, which
triggers the leaf-fall state when the air temperature remains below a certain
temperature threshold for 5-7 consecutive days. This criterion serves to buffer single
extreme events from prematurely triggering canopy changes.
During the defoliation period, the offset counter $t_{offset}$ is decremented at each time
step if $GSI < GSID_{thr}$ until it reaches zero, and then dormancy is triggered;
$$t_{offset} = \begin{cases} t_{offset}^{pre} - f_{day}, & GSI < GSID_{thr} \\ t_{offset}^{pre} & , & GSI \geq GSID_{thr} \end{cases} \qquad (8)$$

2.1.2.2 Growing degree day module
The growing degree day (GDD) model was originated from CLM4. From the



modularization viewpoint, the GDD model is independent of CLM4, so the GDD
model was easy to couple into other LSMs, e.g., the DLM model. Two deciduous
vegetation types are contained in the GDD model. One is seasonally deciduous, and
the other is stress-deciduous. The former refers to the temperate and boreal deciduous
trees; the latter includes temperate and boreal deciduous shrubs, grass and tropical
deciduous trees. The phenophases in this model also contain green-up, normal growth,
defoliation and dormancy, which are assumed to be only driven by climate factors (*e.g.*,
temperature and soil water) and day length.
(1) Seasonal-Deciduous Phenology
Green-up for seasonal-deciduous vegetation is triggered based on an
accumulated GDD approach (White et al. 1997). The GDD summation ($GDD_{sum}$,
degree·day) is initiated at zero when the phenological state is dormant and the model
time step crosses the winter solstice (Oleson et al. 2013). Once the environmental
conditions are met, $GDD_{sum}$ is updated at each time step as:
$$GDD_{sum} = \begin{cases} GDD_{sum}^{pre} + (T_{soil} - 273.15) \cdot f_{day}, & T_{soil} > 273.15 \\ GDD_{sum}^{pre}, & T_{soil} \leq 273.15 \end{cases} \quad (9)$$

where $f_{day} = Vt / 86400$, V$t$ is time step and equals 1800 sec. The superscript *pre*
represents the last time step. $T_{soil}$ is the temperature of the third soil layer in K. When
$GDD_{sum}$ is greater than the GDD summation threshold ($GDD_{thr}$, degree·day), green-up
is triggered, and the onset counter ($t_{onset}$, day) that controls the green-up length is
initialized. The $GDD_{thr}$ is estimated as follows:



$$GDD_{thr} = \exp(4.8 + 0.13 \cdot (T_{air} - 273.15)) \qquad (10)$$

where $T_{air}$ is the annual average air temperature at a 2 m height in degrees K.
During green-up, the onset counter ($t_{onset}$) is decremented at each time step until it
reaches zero, triggering normal growth,
$$t_{onset} = t_{onset}^{pre} - f_{day} \qquad (11)$$

After simulating time past the summer solstice, vegetation defoliation is
triggered if the day length ($DL$, hr) is shorter than the corresponding threshold ($DL_{thr}$,
hr), and the offset counter ($t_{offset}$, day) that controls the defoliation length is initialized
at the beginning of the defoliation period.
During defoliation, the offset counter $t_{offset}$ decreases at each time step until it
reaches zero, triggering dormancy,
$$t_{offset} = t_{offset}^{pre} - f_{day} \qquad (12)$$

(2) Stress-Deciduous Phenology
The process for triggering green-up of stress-deciduous vegetation is more
complex than for the seasonally deciduous vegetation in CLM4. It is influenced by
temperature, soil water and day length simultaneously.
First, the freezing day accumulator for green-up ($FDG_{sum}$, day) is necessary and
is calculated as:
$$FDG_{sum} = \begin{cases} FDG_{sum}^{pre} + f_{day}, & T_{soil} < 273.15 \\ FDG_{sum}^{pre}, & T_{soil} \geq 273.15 \end{cases} \qquad (13)$$

where $f_{day} = \nabla t / 86400$, $\nabla t$ is time step and set to be 1800 sec. The superscript *pre*



represents the last time step. $FDG_{sum}$ is initialized to zero at the beginning of the
dormant period. $T_{soil}$ is the temperature of the third soil layer in K.

If $FDG_{sum} > FDG_{thr}$, where $FDG_{thr}$ is the freezing day summation threshold for

green-up in days, the growing-degree-day summation ($GDD_{sum}$, degree·day) (see Eq.
9) is followed exactly.

Meanwhile, the accumulated soil water index for green-up ($SWIG_{sum}$, days) is

calculated as:

$$SWIG_{sum} = \begin{cases} SWIG_{sum}^{pre} + f_{day}, & \Psi_{soil} \geq \Psi_{onset} \\ SWIG_{sum}^{pre}, & \Psi_{soil} < \Psi_{onset} \end{cases} \tag{14}$$

$SWIG_{sum}$ is initialized to zero at the beginning of a dormant period, $\Psi_{soil}$ is the soil
water potential in the third soil layer in MPa, and $\Psi_{onset}$ is the soil water potential
threshold for green-up in MPa.

Only if $GDD_{sum} > GDD_{thr}$ (or $T_{soil}$ is always greater than 273.15K) and $SWIG_{sum} >$

$SWIG_{thr}$ and $DL > DL_{thr}$ is green-up triggered, where $GDD_{thr}$ is the GDD summation
threshold in degree·days (see Eq. 10), the $SWIG_{thr}$ is the soil water index summation
threshold in days, $DL$ is the day length in hours, and $DL_{thr}$ is the day length threshold
in hours.

At the beginning of the green-up period, an onset counter for controlling the

green-up length ($t_{onset}$, days) is initialized. Then, $t_{onset}$ is decremented at each time step
until it reaches zero, triggering normal growth (see Eq. 11).

During normal growth, any one of the following unfavorable conditions is





sufficient to trigger vegetation defoliation - a sustained period of dry soil, a sustained
period of cold temperature, or a shorter day length.

The dry soil condition is evaluated with the soil water index accumulator for

defoliation ($SWID_{sum}$, day), which is expressed as:

$$SWID_{sum} = \begin{cases} SWID_{sum}^{pre} + f_{day} &, \quad \Psi_{soil} \le \Psi_{offset} \\ \max(SWID_{sum}^{pre} - f_{day}, 0), & \quad \Psi_{soil} > \Psi_{offset} \end{cases} \quad (15)$$

where $\Psi_{offset}$ is the soil water potential threshold for defoliation in MPa.

Meanwhile, the cold temperature condition is calculated with the freezing day

accumulator for defoliation ($FDD_{sum}$, day) and is described as:

$$FDD_{sum} = \begin{cases} FDD_{sum}^{pre} + f_{day} &, \quad T_{soil} \le 273.15 \\ \max(FDD_{sum}^{pre} - f_{day}, 0), & \quad T_{soil} > 273.15 \end{cases} \quad (16)$$

When $SWID_{sum} > SWID_{thd}$ or $FDD_{sum} > FDD_{thr}$ or $DL < DL_{thr}$, defoliation is

triggered. $SWID_{thr}$ is the soil water index summation threshold for defoliation in days
and $FDD_{thr}$ is the freezing day accumulator threshold for defoliation in days.

The offset counter for controlling the defoliation length ($t_{offset}$, days) is initialized

and decreases at each time step until it reaches zero, triggering dormancy (see Eq. 12).
**2.2 Data sets used**
**2.2.1 FLUXNET Data**

The selected 22 eddy-covariance (EC) sites from the FLUXNET database

(http://fluxnet.ornl.gov/) are mainly distributed in North America and Europe (see Fig.
2). The EC sites were selected according to the following requirements: (i) the

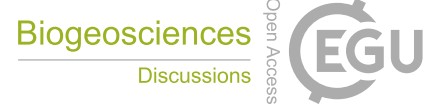

dominant vegetation type at the site was limited to deciduous forest, deciduous shrubs
and grass; (ii) the site provided at least four years of continuous data as a part of a
publicly accessible standardized Level 3 or 4 database; (iii) a 'site-year' was accepted
for analysis if more than 90% of the half hours in a year contained non-missing values
for the meteorological data and the carbon flux data (Chen et al. 2013); and (iv) the
sites represented as many climate zones as possible.
The final selected sites were expected to represent the following four main
climatic environments including temperate, boreal, arid and the moist climate zones
and four biome types containing needleleaf deciduous forests (NDF), broadleaf
deciduous forests (BDF), broadleaf deciduous shrubs (BDS) and grasslands. Different
biome types in a particular climate environment are usually characterized by different
leaf types, leaf longevity and life forms (Roth et al. 2015). Thus, a biome type located
in a particular climate zone can represent the corresponding plant function type (PFT).
A description of the information for the selected sites classified by PFT can be found
in Table 2.
Every site contained half-hourly meteorological and GPP data for 4 consecutive
years. The data for the first two consecutive years were used to optimize the model
parameters and for the next two consecutive years to evaluate the simulation results of
DLM. Meteorological data including down-welling solar radiation, precipitation, wind
speed, air temperature and relative humidity were applied to drive DLM. The
EC-measured GPP data were used for model calibration and assessment.



**2.2.2 GPP and phenology data**

GPP data usually have gaps. If the gaps were less than 2 hours, the missing

values were filled by piecewise linear interpolation. To fill longer gaps, the light use
efficiency (LUE) model was employed (Monteith 1972; Sims et al. 2008). Though
other uncertainties still existed in the EC-measured GPP, *e.g.*, underestimation of the
ecosystem respiration at night (Schaefer et al. 2012), they were still regarded as the
'ground truth' in this study.

The phenological observations used for evaluating the simulated phenophases of

DLM contained two parts. One part was derived from the EC-measured GPP, and the
other was derived from the observed green chromatic coordinate (GCC) data.

The phenological inversion method based on the GPP data used the ratio of the

daily GPP to the growing season amplitude to identify the phenophases (Melaas et al.
2013), which only retrieved the start of the growing season and the end of the growing
season.

The GCC data were derived from the digital images photographed by an

automated and high-frequency digital camera that is generally applied in modern
phenological observation (Ide; Oguma 2010) and were calculated from the average
red (R), green (G), and blue (B) pixel digital numbers (DNs) over the region of
interest (ROI), *i.e.*, GCC = G/(G+R+B) (Ahrends et al. 2008; Ahrends et al. 2009;
Sonnentag et al. 2012). Quality control of the GCC data was necessary to correct for





gaps and false data before using a smoothed curve for fitting, following the approach
of Ludvig *et al.*(Ludvig 2014). The inflection points of the curve were calculated to
identify the phenophases. The general smoothed curves contained the loGSItic
function, the double-loGSItic function, the Asymmetric Gaussian function, *etc*. (Ide;
Oguma 2010; Klosterman et al. 2014). Some studies indicated the Scurve function
describing the vegetation growing state better than the loGSItic function and the
Asymmetric Gaussian function (Che et al. 2014a; Che et al. 2014b). Thus, the Scurve
function was used here to fit the GCC data, and the corresponding process for
extracting phenophases based on the Scurve function was carried out. The final
inversion phenophases included the start of the growing season, normal growth,
defoliation and the end of the growing season. Simultaneously, visual interpretation of
the digital images was also used to appropriately correct the retrieved results. The
digital images were downloaded from the PhenoCam Network
(http://phenocam.sr.unh.edu/webcam/). Considering the geographic position and the
site-year of the flux sites, after selection, the PhenoCam sites only contained the
US-MOz site (*i.e.*, the Columbiamissouri site). Fig. 3 shows the digital images for key
phenophases at this site. The plants started to green-up in early April (Fig. 3a) and
entered into normal growth in the middle of May (Fig. 3b). Leaves began to fall
widely in later October (Fig. 3c), and dormancy began in the middle of November
(Fig. 3d).
Admittedly, certain uncertainties existed in the two phenological observations.



For example, the retrieval phenophases of the GPP were deeply affected by the quality
itself. The GPP ratio method was a dynamic threshold method. Before using it, the
GPP data were first smoothed by the cubic spline. Even so, this method was still
sensitive to high GPP values occurring in early spring or later autumn. If the GPP high
values were noisy, the retrieval phenophases would have large uncertainties. The GCC
data might be distorted at a certain time due to the effect of camera firmware, the
white balance setting, changes in illumination and smog, etc. (Ahrends et al. 2008; Ide;
Oguma 2010). Furthermore, delimiting the ROI of the image and using the phenology
inversion method might affect the accuracy of the phenological inversion results
(Ahrends et al. 2009; Klosterman et al. 2014). However, if the ground measured
phenological observations were absent, the retrieval phenophases based on the GPP
and GCC data were still considered as the 'observed values' for model evaluation.
**2.3 Model suns and parameters optimization**
**2.3.1 Model control tests and runs**

To evaluate the performance of the two alternative phenology models coupled to

DLM, a control test was designed. Based on DLM, two versions of the model were
built by coupling the GSI and GDD models, respectively, which are designated as
DLM-GSI and DLM-GDD.

Through the control test, the accuracy of simulating phenophases using the two

versions of DLM can be objectively assessed. Additionally, the effects of the

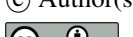



phenology models on GPP simulated using the two versions can be evaluated.
We ran separately the two DLM versions (DLM-GSI and DLM-GDD) at
half-hourly time steps with the same observed meteorological and land surface data as
inputs. Missing meteorological data were supplied by linear interpolation for gaps of
less than 2 hours (Chen et al. 2013). Various methods were used for filling longer gaps
for different variables. For variation trends of the down-welling solar radiation and air
temperature, the sine function was appropriate. For relative humidity, the cosine
function was suitable. Considering their strong randomness, the piecewise linear
interpolation approach was used for precipitation and wind speed.
The soil texture (*i.e.*, percentages of sand and clay) was obtained from the site
information or published articles. Other soil property data were obtained from
CESM1.0.3 as a source of land surface data for the year 2000 (Lawrence et al. 2011).
The soil state variables (*e.g.*, soil temperature and moisture) and vegetation state
variables (*e.g.*, LAI, stem area index (SAI) and canopy top and bottom heights) at
each site for the off-line simulations were obtained from the initialization. The
initialization was acquired from a long (at least 2000 years) spin-up simulation until
the carbon and nitrogen pools and associated LAI, SAI, and vegetation heights
approximated the equilibrium with the repeating atmospheric forcing data for the
years of 1972-2001 (Qian et al. 2006) provided by NCAR. The $CO_2$ concentration,
nitrogen and aerosol deposition at year 2000 levels at each site were also provided by
NCAR.



**2.3.2 Parameters optimization**
The PFT-dependent parameters for vegetation physiology, *e.g.,* the leaf
maximum carboxylation rate at 25 °C and the leaf stomatal
resistance-to-photosynthesis relationship in DLM, were slightly adjusted based on
published parameters (Chen et al. 2013). The foliage clumping index in DLM was
taken from published papers (Chen et al. 2007; Chen et al. 2013).
The parameters in the GSI phenological modules were initialized by referring to
literatures (Jolly et al. 2005; Stöckli et al. 2008). These phenological parameters were
further optimized based on EC-measured GPP using the simulated annealing (SA)
algorithm (Dong et al. 2013; Li et al. 2004), which was not only independent of the
cost function but also able to produce global optimal parameters of the model. The
final optimized parameters of the GSI model can be found in Table 3.
The parameters in the GDD phenological model were designed to be independent
of the PFTs and originated from the CLM4 technical manual (Oleson et al. 2013;
Oleson et al. 2010). The final parameters are as follows: $N_{onset}$ = 30 day, $N_{offset}$ = 15 day,
$FDG_{thr}$ = 15 day, $FDD_{thr}$ = 15 day, $\Psi_{onset}$ = -2 MPa, $\Psi_{offset}$ = -2 MPa, $SWIG_{thr}$ = 15
day, $SWID_{thr}$ = 15 day, $DL_{thr}$ = 11 hr, where $N_{onset}$ is the initialized onset counter for
controlling the length of green-up, $N_{offset}$ is the initialized offset counter for controlling
the length of defoliation, $FDG_{thr}$ is the freezing day summation threshold for green-up,
$FDD_{thr}$ is the freezing day summation threshold for defoliation, $\Psi_{onset}$ is the soil




water potential threshold for green-up, $\Psi_{offset}$ is the soil water potential threshold for
defoliation, $SWIG_{thr}$ is the soil water index summation threshold for green-up, $SWID_{thr}$
is the soil water index summation threshold for defoliation, and $DL_{thr}$ is the day length
threshold.
**2.4 Model evaluation methods**
For assessing the model performance, statistical analyses containing bias (Eq.
17), absolute bias (Eq. 18), root mean square error (RMSE, Eq. 18) and index of
agreement (IA, Eq. 19) were used (Willmott 1982).

$$Bias = \frac{1}{n}\sum_{i=1}^{n}(P_i - O_i) , \tag{17}$$

$$ABias = \frac{1}{n}\sum_{i=1}^{n}|P_i - O_i| , \tag{18}$$

$$RMSE = \sqrt{\frac{1}{n}\sum_{i=1}^{n}(P_i - O_i)^2} , \tag{19}$$

$$IA = 1 - \frac{\sum_{i=1}^{n}(P_i - O_i)^2}{\sum_{i=1}^{n}(|P_i - \overline{O}| + |O_i - \overline{O}|)^2} \tag{20}$$

where $P$ is the model simulated value, $O$ is the observed value, $\overline{O}$ is the observed
mean, and $i$ and $n$ represent the sequence number and the total number of data points,
respectively.





## 3 Results

### 3.1 Simulation of phenological events



First, the simulated phenophases using DLM-GSI and DLM-GDD were
compared with observations derived from the GCC data at the US-MOz site (Fig. 4).
A comparison of corresponding phenological absolute biases (Abias) can be found in
Fig. 5. Both of two versions of DLM simulated the phenophases well at this site.
However, the differences in the simulated phenophases were also evident.
The simulated start of growing season derived from DLM-GSI and DLM-GDD
were earlier and later than the observed values, respectively. The Abias of the
DLM-GSI was 3 days less than that of DLM-GDD on average. The difference
between the simulated normal growth phenophases using the two versions of DLM
was also obvious. The DLM-GSI estimated the phenophase earlier, but the
DLM-GDD estimated it later. The Abias of the former was 4 days less than that of the
latter on average. For defoliation, Both DLM-GSI and DLM-GDD estimated the
phenophase earlier, but the former had a lower prior-estimation error (Abias = 4days)
than the latter (Abias = 8days). For the EGS simulation, the results of DLM-GSI and
DLM-GDD were later and earlier than the observed values, respectively, and the EGS
Abias of DLM-GSI was 5 days less than DLM-GDD.
The above analysis indicates the simulated phenophases of DLM-GSI were much
closer to observed values than those of DLM-GDD, and the DLM-GSI estimated SGS





and EGS earlier and later, respectively, but DLM-GDD did the opposite.

The simulation performance of two versions of DLM was assessed by using

observations derived from the EC-measured GPP at all sites. A comparison of the
phenophases simulated by the two versions of DLM and the observed values is shown
in Fig. 6. In this study, we focused on the start of the growing season (Fig. 6a) and the
end of the growing season (Fig. 6b) at the EC sites. A corresponding comparison of
the absolute biases for the simulated phenophases is shown in Fig. 7.

As shown in Figs. 6 & 7, the differences between the phenophases simulated by

the two versions of DLM were remarkable, and the differences also existed for each
plant function type. In Fig. 8, the boxplot shows the discrete character of the absolute
biases for the simulated results by using the two versions for each PFT. For boreal
needleleaf deciduous forest (BNDF) (Figs. $8a_1$ & $8b_1$), the Abias range and
interquartile range of the simulated SGS using DLM-GSI were both lower than those
simulated using DLM-GDD, as were the mean and median of the SGS Abiases. The
Abias range, mean and median of the simulated EGS using DLM-GSI were all lower
than those of DLM-GDD, but the Abias interquartile range was higher. At the BNDF
sites, the accuracy of the phenophases simulated using DLM-GSI at the CA-NS1 site
and the FI-Hyy site was obviously higher than those simulated using DLM-GDD. The
results showed that the GSI model reduced the SGS and EGS Abiases of DLM at the
CA-NS1 site by 6 and 30 days, respectively. As the same time, the GSI model reduced
the SGS and EGS Abiases of DLM at the FI-Hyy site by 29 days and 8 days,



respectively.

For temperate broadleaf deciduous forest (TBDF) (Figs. $8a_2$ & $8b_2$), the Abias

range and interquartile range of the SGS simulated by DLM-GSI were both shorter
than those of DLM-GDD, as were the mean and median of SGS Abiases. The Abias
range of EGS simulated by DLM-GSI was consistent with that simulated by
DLM-GDD. The Abias mean and median of simulated EGS using DLM-GSI were
slightly lower than the values obtained using DLM-GDD, but the interquartile range
was higher for DLM-GSI compared with DLM-GDD. At the TBDF sites, the
simulated results using DLM-GSI at the CH-Lae site and the US-MOz site were much
closer to observed values than using DLM-GDD. The results showed that the GSI
model reduced the SGS and EGS Abiases of DLM at the CH-Lae site by 32 days and
21 days, respectively. At the same time, the accuracy of simulated SGS using
DLM-GSI at the FR-Fon site and the IT-Col site was also higher than that of using
DLM-GDD. However, the accuracy of simulated EGS using DLM-GSI was lower
than that of using DLM-GDD. At the US-Los site, the accuracy of simulated
phenophases using DLM-GSI was inferior to DLM-GDD.

For the boreal broadleaf deciduous forest (BBDF) (Figs. $8a_3$ & $8b_3$), the Abias

range and interquartile range of simulated SGS using DLM-GSI were both less than
using DLM-GDD, as were the mean and median of SGS Abiases. The Abias range,
mean and median of simulated EGS uisng DLM-GSI were all lower than using
DLM-GDD, but the Abias interquartile range was higher for DLM-GSI compared with



DLM-GDD. At the BBDF sites, the accuracies of simulated phenophases using
DLM-GSI exceeded those of using DLM-GDD largely, especially for the DE-Gri site,
the DK-Sor site and the BE-Vie site. The results showed that the GSI model reduced
the SGS and EGS Abiases uisng DLM at the DE-Gri site by 28 and 7 days,
respectively.

For the temperate and boreal broadleaf deciduous shrubs (BDS) (Figs. 8a$_4$, 8b$_4$,

8a$_5$ & 8b$_5$), the Abias range and interquartile range of simulated SGS and EGS using
DLM-GSI were all lower than those using DLM-GDD, as were the Abias mean and
median. At the BDS sites, the accuracy of simulated phenophases using DLM-GSI
was higher than using DLM-GDD widely, especially for the US-Fwf site and the
CA-NS6 site. The results showed that the GSI model reduced the SGS and EGS
Abiases of DLM at the CA-NS6 site by 17 and 58 days, respectively. At the US-Ivo
site, the simulated phenophases using DLM-GSI were consistent with using
DLM-GDD.

For temperate grass (Figs. 8a$_6$ & 8b$_6$), the Abias range of modeled SGS using the

two versions of DLM were both broad, but the Abias interquartile range, mean and
median of simulated SGS using DLM-GSI were all shorter than using DLM-GDD.
However, the Abias range and interquartile range of simulated EGS using DLM-GSI
were both narrower than using DLM-GDD, as were the EGS Abias mean and median.
Compared to the general accuracy of simulated phenophases using both two versions
of DLM for all sites (Figs. 8a$_7$ & 8b$_7$), the phenological Abias range and interquartile





range of using DLM-GSI were both shorter than using DLM-GDD, as were the Abias
mean and median. At the grass sites, the phenological accuracy of the DLM-GSI was
generally higher than that of using DLM-GDD. Nevertheless, the GSI model
indistinctively increased the EGS accuracy of using DLM at the PT-Mi2 site and
US-Wkg site.

The above analysis indicates that the Abias range and interquartile range of using

DLM-GSI were both shorter, and the Abias mean and median were both lower,
showing that the simulated results of DLM-GSI were more stable and reasonable than
those using DLM-GDD. The GSI model significantly decreased the Abias of the
phenophases simulated by the DLM compared to using the GDD model. By using the
GSI model, the Abias of SGS simulated using DLM decreased by 48.2% on average
while the Abias of EGS declined by 39.6%.
**3.2 GPP simulations**

A comparison of simulated GPP using DLM-GSI and DLM-GDD with the

observed values is shown in Fig. 9. The corresponding root mean square errors
(RMSEs) and indices of agreement (IA) for GPP simulation are shown in Fig. 10. By
adopting different phenology models under conditions for which the phenophase
could be estimated, DLM can simulate daily GPP well. The simulated GPP using
DLM-GSI was consistent with DLM-GDD. However, the differences between
simulated GPP were also quite obvious for each PFT and at each site.





Table 4 shows RMSE and IA of simulated GPP using the two versions of DLM
for different PFTs. Obviously, DLM-GSI had lower RMSEs and higher IAs compared
to DLM-GDD for all PFTs. For the PFTs of the TBDS, the BBDS and the temperate
grass, the GPP RMSE of using DLM-GSI was lower than using DLM-GDD by at
least 15%. The GPP IA of using DLM-GSI was higher than using DLM-GDD by at
least 12%. The GSI model sharply improved the accuracy of simulated GPP by using
DLM for these PFTs. For the PFT of BNDF, the GPP RMSE of using DLM-GSI was
lower thanusing DLM-GDD by 6.4%, and the GPP IA exceeded it by 3.9%. The GSI
model clearly improved the accuracy of simulated GPP by using DLM. For the PFTs
of TBDF and the BBDF, the GSI model slightly improved the accuracy of simulated
GPP with DLM compared to using GDD model, decreasing the GPP RMSE of uisng
DLM by only 2.0% - 3.5% and increasing the corresponding IA by only 0.4% - 1.8%.
At the BNDF sites, the GSI model sharply improved the accuracy of simulated
spring GPP using DLM at the CA-NS1 site and the FI-Hyy site and also obviously
improved the accuracy of simulated autumn GPP using DLM at the CA-NS1 site. The
results showed the GSI model reduced the simulated GPP RMSE of using DLM in
spring at the FI-Hyy site by 36.5% and increased the corresponding IA by 75.9%. At
the TBDF sites, the GSI model significantly improved the accuracy of simulated
spring GPP using DLM at the CH-Lae site. The GSI model decreased the GPP RMSE
of using DLM in spring at this site by 19.1% and increased the corresponding IA by
20.2%. For the other TBDF sites, a lesser improvement of simulated GPP accuracy by





535 the GSI model using DLM in the growing season was noted. At some sites, the

536 accuracy of simulated GPP based on the GSI model was lower than for the GDD

537 model. At the BBDF sites, the GSI model sharply improved simulated GPP accuracy

538 of using DLM at the DK-Sor site, the BE-Vie site and DE-Gri site. The GSI model

539 reduced the GPP RMSE of using DLM in spring at the DK-Sor site by 29.5% and

540 increased the corresponding IA by 85.0%. The GSI model also decreased the autumn

541 GPP RMSE of using DLM at this site by 7.5% and increased the corresponding IA by

542 4.3%. At the DE-Hai site, the estimated SGS and EGS using DLM-GSI was

543 respectively earlier and later compared to the observed values. The Abiases for the

544 SGS and EGS of using DLM-GSI were both higher than using DLM-GDD. Thus, the

545 GPP results simulated using DLM-GSI were inferior to DLM-GDD at this site. At the

546 TBDS sites, the GSI model significantly improved the accuracy of simulated GPP

547 using DLM at the CA-Mer site and the US-Fwf site. Meanwhile, the GSI model

548 obviously improved the accuracy of simulated spring GPP using DLM at the US-Ton

549 site. The results showed the RMSE of simulated spring GPP using DLM-GSI at the

550 CA-Mer site was lower than using DLM-GDD by 17.5%, and the corresponding IA

551 was higher by 20.5%. The RMSE of simulated autumn GPP using DLM-GSI at this

552 site was lower than using DLM-GDD by 3.8%, and the corresponding IA was higher

553 by 4.1%. At the BBDS sites, the GSI model significantly improved the accuracy of

554 GPP simulated using DLM at the CA-NS6 site and the US-Ivo site. At the temperate

555 grass sites, the GSI model also significantly improved the accuracy of GPP simulated





using DLM at most sites.

From the above analysis, the GSI model significantly improved the accuracy of

simulated GPP in DLM for different PFTs compared to the GDD model. For most of
the sites, the RMSEs of simulated GPP using DLM-GSI were lower than using the
DLM-GDD model, and the IA was on the contrary, especially for GPP simulation in
spring and autumn. Over all, the GSI model increased the accuracy of GPP simulation
by using DLM compared to using the GDD model. The GSI model reduced the GPP
RMSE of using DLM by 8.0%, and increased the corresponding IA by 7.5%.

## 4 Discussions

According to the characteristics of climate zones, the sites can be divided into a

moist climate zone and an arid climate zone. Summarizing accuracies of simulated
phenophases for these two kinds of sites showed that the Abias range and interquartile
range of the phenophases simulated using DLM-GSI and DLM-GDD for the moist
climate sites were less broad than those for the arid climate sites, as were the Abias
mean and median. For example, the Abias interquartiles for the SGS simulated using
DLM-GSI for the moist and arid climate sites were 18 and 24 days, respectively, and
the Abias interquartiles for the EGS simulated using DLM-GSI for the moist and arid
climate sites were 10 and 15 days, respectively. Meanwhile, the Abias interquartiles
for the SGS simulated using DLM-GDD for the moist and arid climate sites were 22
and 59 days, respectively, and the Abias interquartiles for the EGS simulated using




DLM-GDD at the moist and arid climate sites were 10 and 27 days, respectively. Thus,
the accuracies of the phenophases simulated with the phenology models for the moist
climate sites were higher than for the arid climate sites. At the temperate arid sites, the
effect of moisture on the vegetation phenology is second important compared to that
of temperature. In the warm temperate arid sites, the importance of water was even
greater than that of temperature. Fig. 11 shows the effect of the sensitivities of the
phenology parameters on the growing season index at the US-Wkg site. The
sensitivities of temperature and vapor pressure deficit were both important to the
growing season index. However, the effect of the temperature sensitivity (see the error
bars in light red in Fig. 11) on the growing season index was confined to the outside
of the growing season (see the green dashed line derived from the EC-measured GPP
in Fig. 11). The effect of VPD sensitivity (see the error bars in light blue in Fig. 11) on
the growing season index was mainly located in the growing season. That is to say, at
the US-Wkg site, the adjustment of temperature parameters made little contribution to
improving the accuracy of the phenophases simulated by the GSI model, but the
adjustment of VPD parameters was on the contrary. Nevertheless, the accurate
acquisition of VPD parameters at this site was not easy. In addition, the parameters
used in this study for simulating the phenophases were calculated from the average
parameters at different sites for which the PFT was the same. Even if the precise VPD
parameters could be obtained at this site, the uncertainty was still large when the
values were averaged.

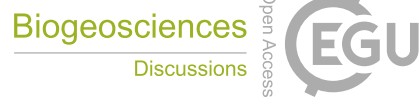

Furthermore, the GPP observations at the US-Wkg site (Fig. 9v) indicated that
the growing seasons were bimodal. The VPD parameters and the threshold parameters
for triggering the phenophases used in the GSI model were all constant. This scheme
could lead to a certain bias when the GSI model was used to simulate phenophases at
the sites at which the number of growing seasons was greater than one. This also
occurred at the temperate arid sites, such as the PT-Mi2 and the US-Ton sites. The
statistics showed that the accuracy of the phenophases simulated with DLM-GSI and
DLM-GDD at the single-season vegetation sites was higher than at the multi-season
vegetation sites.
For example, the bias of phenophases simulated using DLM-GDD at the
US-Wkg site was large. The annual average air temperature was approximately 17.25
ºC at the US-Wkg site, and the annual minimum temperature (-3 ºC) occurred in
winter. Similar to the GSI model, the effect of temperature on triggering the
phenophases for the GDD model was weak at this site. The annual precipitation was
approximately 245.78 mm at this site. The sparse precipitation was the main factor
controlling the vegetation phenology. The GDD model estimated the SGS and the
EGS by calculating the cumulative days when the soil water potential (SWP) was
higher or lower than -2 MPa, but the starting dates when the SWP estimated using
DLM-GDD continuously exceeded -2 MPa in 2006 were April 19 and July 20. In fact,
the simulated SWP using DLM was inconsistent with the observed values for both
days, causing large biases in the phenophases simulated using DLM-GDD compared





to the observed values. The SWP variable was a derivative in DLM. For that reason,
the adoption of the derivative variables by the GDD model to simulate the
phenophases was not ideal. Similar to the GSI model, the threshold parameters (*e.g.,*
the threshold of SWP) in the GDD model were constant and were also deficient for
phenophase simulation at the multi-season vegetation sites. The defective model
structure and uncertainty in parameters caused the simulated phenophases using the
GDD model to have large biases at other sites (*e.g.,* the PT-Mi2 site and the US-FPe
site).

Compared to the observed values, the Abiases of simulated phenophases using

the two versions of DLM were significant, although the Abiases of using DLM-GSI
were comparatively less, indicating that the two phenology models still must be
further developed and perfected by future studies. In addition, the DLM must also be
improved, particularly by obtaining more accurate simulated variables as inputs for
the phenology models.
**5 Conclusion**

The two different phenological schemes, the GSI and the GDD models, were

coupled to DLM and were evaluated for deciduous forests and grasses against the
observed phenology. Through control tests, the simulated phenophases and GPP by
the two versions of DLM were analyzed and compared. The main conclusions are as
follows:





(i) Compared with the phenological observations derived from the GCC data at
the US-MOz site, DLM-GSI had lower absolute biases for estimating the phenophases
including the start of the growing season, normal growth, defoliation and the end of
the growing season compared to DLM-GDD. The simulated phenophases using
DLM-GSI were much closer to the observed values than those using DLM-GDD at
this site. The start of the growing season was estimated earlier using DLM-GSI but
later using DLM-GDD at the US-MOz site. Meanwhile, the end of growing season
was estimated later using DLM-GSI but earlier using DLM-GDD.
(ii) By comparing against the phenological observations derived from the GPP
data at all sites, the absolute bias of the phenophases simulated using DLM-GSI had a
tighter range and interquartile range than using DLM-GDD and a lower mean and
median than using DLM-GDD for various PFTs, indicating that the simulated results
of using DLM-GSI were more stable and reasonable than using DLM-GDD. Overall,
the GSI model significantly decreased the absolute bias of the phenophases simulated
using DLM at all sites compared to the GDD model. Additionally, the use of the GSI
model decreased the absolute bias of the SGS simulated using DLM by 48.2% on
average and the absolute bias of the EGS declined by 39%.
(iii) The GSI model significantly improved the accuracy of the GPP simulated
using DLM compared to the GDD model for various PFTs. For most of the sites, the
RMSE of simulated GPP using DLM-GSI was lower than that of using DLM-GDD,
and the IA was higher using DLM-GSI than using DLM-GDD, especially for GPP



simulation in spring and autumn. Over all, the GSI model improved the accuracy of
GPP simulation using DLM compared with using the GDD model at all sites. The GSI
model reduced the simulated GPP RMSE of the DLM model by 8.0% and increased
the corresponding IA by 7.5%.

**Acknowledgements.** This research was financially supported by a research grant

(2013BAC03B04) of National Key Technology Support Program and a research grant
(41271116) funded by the National Science Foundation of China, a research grant
(2012ZD010) of the Key Project for the Strategic Science Plan in Institute of
Geographic Sciences and Natural Resources Research, Chinese Academy of Sciences
(CAS), and Jiangsu Provincial 'double creation' program. We acknowledge the
agencies that supported the operations at the flux towers used here, which are parts of
FLUXNET (http://fluxnet.ornl.gov/). We thank the PhenoCam Network for making
the phenology imagery freely available for download (http://phenocam.sr.unh.edu/).
We thank Dr. Huifang Zhang, Dr. Jing Chen, Dr. Xianming Dou, Ms.Yuchen Wang,
Ms. Xiaofeng Lin and Mr. Shaobo Sun of IGSNRR for downloading and gap-filling
FLUXNET data.





**Tables caption**
**Table 1. An algorithmic comparison among EASS, CLM4 and DLM.**
**Table 2. Descriptions of global FLUXNET sites used.**
**Table 3. Phenological parameters in the DLM-GSI model.**
**Table 4. Comparison of the root mean square error (RMSE) and the index of**
**agreements (IAs) for gross primary production simulation for different**
**vegetation types using the DLM-GSI and DLM-GDD models.**






**Table 1. An algorithmic comparison among EASS, CLM4 and DLM.**

| Algorithms | EASS | CLM4 | DLM |
|---|---|---|---|
| Canopy layers | Two layers (overstory and understory) | One layer | Two layers (overstory and understory) |
| Snow layers | depending on snow depth | depending on snow depth | depending on snow depth |
| Soil layers | 7 | 15 | 15 |
| Canopy up-scaling | two-leaf strategy | one-leaf strategy | two-leaf strategy |
| Two-leaf (sunlit and shaded leaves) strategy implementation | based on fractions of sunlit and shaded leaves at a canopy depth as described by Dai et al.,(2004), also depending on the clumping index related of PFTs | based on fractions of sunlit and shaded leaves at a canopy depth as described by Dai et al.,(2004) | based on fractions of sunlit and shaded leaves at a canopy depth as described by Dai et al., (2004), also depending on the clumping index related of PFTs |
| Photosynthesis | two-leaf strategy, Rubisco-limited rate and light-limited rate are both based on Chen et al., (1999) and Wang and Leuning, (1998) | two-leaf strategy, Rubisco-limited rate and light-limited rate are both based on Bonan et al.,(2011) | two-leaf strategy, Rubisco-limited rate and light-limited rate are both based on Chen et al., (1999) and Wang and Leuning, (1998) |
| Evapo-transpiration | two-leaf strategy, Penman-Monteith equation | one-leaf strategy, Mass-transfer equation | two-leaf strategy, Penman-Monteith equation |
| Land cover type | 6 vegetation types, burned area, barren land, urban area and permanent snow/ice area | 15 possible PFTs, bare ground, crop, lake, urban and glacier | 15 possible PFTs, bare ground, crop, lake, urban and glacier |
| Phenology | derived from leaf area index (LAI) | growing degree days (GDDs) model accompanying with day length and soil moisture restriction | growing season index(GSI) model |
| Vegetation carbon pools | as a whole | leaf, live stem, dead stem, live coarse root, dead coarse root, fine root, storage organs and respiration organs | leaf, live stem, dead stem, live coarse root, dead coarse root, fine root, storage organs and respiration organs |
| Litter carbon pools | Coarse detritus from woody and coarse root, surface structural litter, surface metabolic litter, surface microbe pool | coarse woody debris (CWD), 3 litter pools | coarse woody debris (CWD), 3 litter pools |





| | | | |
|---|---|---|---|
| Soil carbon pools | soil structural litter pool, soil metabolic pool, soil microbe pool, slow carbon pool, passive carbon pool | 4 soil organic matter pools | 4 soil organic matter pools |
| Reference | (Chen et al. 2007; Chen et al. 1999; Dai et al. 2004; Wang; Leuning 1998) | (Bonan et al. 2011; Dai et al. 2004; Lawrence et al. 2011; Oleson et al. 2013; Oleson 2010; Thornton; Zimmermann 2007; Thornton et al. 2002; White et al. 1997) | (Chen et al. 2007; Chen et al. 2013; Chen et al. 1999; Dai et al. 2004; Jolly et al. 2005; Oleson et al. 2013; Oleson 2010; Stöckli et al. 2008; Wang; Leuning 1998) |





**Table 2. Descriptions of global FLUXNET sites used.**

| NO. | Site ID[a] | Lon. (˚E) | Lat. (˚N) | Elev. (m) | Biome Type[b] | Climate Zone | Air Temp. (°C yr⁻¹) | Percip. (mm yr⁻¹) | SiteYear |
|---|---|---|---|---|---|---|---|---|---|
| 1 | CA-NS1 | -98.48 | 55.88 | 253 | NDF | Boreal (Moist) | 0.59 | 201.40 | 2002-2005 |
| 2 | CA-Oas | -106.20 | 53.63 | 580 | NDF | Boreal (Moist) | 1.95 | 541.75 | 2003-2006 |
| 3 | FI-Hyy | 24.30 | 61.85 | 185 | NDF | Boreal (Moist) | 4.59 | 499.08 | 2004-2007 |
| 4 | CH-Lae | 8.37 | 47.48 | 689 | BDF | Cool Temperate (Moist) | 7.73 | 846.40 | 2005-2006, 2008-2009 |
| 5 | FR-Fon | 2.78 | 48.48 | 100 | BDF | Warm Temperate (Dry) | 11.35 | 668.08 | 2005-2008 |
| 6 | IT-Col | 13.59 | 41.85 | 1560 | BDF | Warm Temperate (Moist) | 7.44 | 994.04 | 2003-2006 |
| 7 | US-Los | -89.98 | 46.08 | 485 | BDF | Cool Temperate (Moist) | 5.10 | 694.82 | 2001-2004 |
| 8 | US-MOz | -92.20 | 38.74 | 212 | BDF | Warm Temperate (Moist) | 14.00 | 699.00 | 2004-2007 |
| 9 | BE-Vie | 6.00 | 50.31 | 450 | BDF | Boreal (Moist) | 8.36 | 1070.09 | 2005-2008 |
| 10 | DE-Gri | 13.51 | 50.95 | 385 | BDF | Boreal (Moist) | 8.72 | 874.33 | 2005-2008 |
| 11 | DE-Hai | 10.45 | 51.08 | 430 | BDF | Boreal (Moist) | 8.23 | 801.50 | 2004-2007 |
| 12 | DK-Sor | 11.65 | 55.49 | 40 | BDF | Boreal (Moist) | 8.54 | 658.86 | 2003-2006 |
| 13 | CA-Mer | -75.52 | 45.41 | 65 | BDS | Cool Temperate (Moist) | 6.26 | 1048.18 | 2005-2008 |
| 14 | US-Fwf | -111.77 | 35.45 | 2316 | BDS | Cool Temperate (Dry) | 8.63 | 895.78 | 2005-2008 |
| 15 | US-Ton | -120.97 | 38.43 | 170 | BDS | Warm Temperate (Dry) | 16.32 | 535.86 | 2002-2003, 2006-2007 |
| 16 | CA-NS6 | -98.96 | 55.92 | 271 | BDS | Boreal (Moist) | -0.86 | 256.05 | 2002-2005 |
| 17 | US-Ivo | -155.75 | 68.49 | 557 | BDS | Boreal (Moist) | -9.11 | 292.99 | 2003-2006 |
| 18 | AT-Neu | 11.32 | 47.12 | 970 | GRA | Cool Temperate (Moist) | 6.52 | 718.35 | 2003-2006 |
| 19 | FI-Kaa | 27.30 | 69.14 | 155 | GRA | Cool Temperate (Moist) | 0.46 | 459.73 | 2000-2001, 2004-2005 |
| 20 | PT-Mi2 | -8.03 | 38.48 | 190 | GRA | Warm Temperate (Dry) | 14.21 | 575.69 | 2005-2008 |
| 21 | US-FPe | -105.10 | 48.31 | 638 | GRA | Cool Temperate (Dry) | 5.79 | 428.60 | 2003-2006 |
| 22 | US-Wkg | -109.94 | 31.74 | 1524 | GRA | Warm Temperate (Dry) | 17.25 | 245.78 | 2004-2007 |

[a] The site ID is taken from FLUXNET.
[b] Biome types: needleleaf deciduous forest (NDF), broadleaf deciduous forest (BDF),
broadleaf deciduous shrub (BDS), and grassland (GRA).



**Table 3. Phenological parameters in the DLM-GSI model[a]**

| Biome type | Climate zone | $DL_{max}$ (hr) | $DL_{min}$ (hr) | $T_{max}$ (K) | $T_{min}$ (K) | $VPD_{max}$ (Pa) | $VPD_{min}$ (Pa) | $GSIG_{thr}$ - | $GSID_{thr}$ - | $N_{onset}$ (day) | $N_{offset}$ (day) |
|---|---|---|---|---|---|---|---|---|---|---|---|
| NDF | Boreal | 11.50 | 10.75 | 273 | 267 | 2113 | 886 | 0.5 | 0.5 | 37 | 32 |
| BDF | Temperate | 11.50 | 10.50 | 280 | 277 | 3084 | 899 | 0.5 | 0.5 | 31 | 32 |
| BDF | Boreal | 11.50 | 10.50 | 282 | 270 | 2095 | 916 | 0.5 | 0.5 | 36 | 17 |
| BDS | Temperate | 11.25 | 9.25 | 276 | 272 | 3199 | 912 | 0.5 | 0.5 | 27 | 28 |
| BDS | Boreal | 11.50 | 10.50 | 281 | 270 | 2100 | 903 | 0.5 | 0.5 | 32 | 31 |
| GRA(C3) | Temperate | 10.25 | 9.25 | 278 | 268 | 2270 | 700 | 0.5 | 0.5 | 27 | 30 |
| Average[b] | | 11.25 | 10.13 | 278 | 271 | 2477 | 869 | 0.5 | 0.5 | 31 | 28 |

[a]parameters: the maximum day length threshold ($DL_{max}$), the minimum day length
threshold ($DL_{min}$), the maximum air temperature threshold ($T_{max}$), the minimum air
temperature threshold ($T_{min}$), the maximum vapor pressure deficit threshold ($VPD_{max}$),
the minimum vapor pressure deficit threshold ($VPD_{min}$), the threshold for triggering
the vegetation green-up ($GSIG_{thr}$), the threshold for triggering the vegetation
defoliation ($GSID_{thr}$), the initialized onset counters for controlling the green-up length
($N_{onset}$), and the initialized offset counters for controlling the defoliation length
($N_{offset}$).
[b]Average was calculated for all biome types and climate zones.



**Table 4. Comparison of the root mean square error (RMSE) and the index of**
**agreements (IAs) for gross primary production simulation for different**
**vegetation types using the DLM-GSI and DLM-GDD models.**

| Biome type Climate zone | RMSE (gC m$^{-2}$ d$^{-1}$) | | IA | |
|---|---|---|---|---|
| | DLM-GSI | DLM-GDD | DLM-GSI | DLM-GDD |
| NDF Boreal | 2.055 | 2.197 | 0.830 | 0.799 |
| BDF Temp. | 2.759 | 2.817 | 0.842 | 0.838 |
| BDF Boreal | 3.399 | 3.523 | 0.846 | 0.830 |
| BDS Temp. | 1.420 | 1.689 | 0.786 | 0.696 |
| BDS Boreal | 0.764 | 1.035 | 0.858 | 0.742 |
| GRA Temp. | 1.349 | 1.642 | 0.733 | 0.619 |
| Average | 2.095 | 2.278 | 0.810 | 0.753 |




**Figures caption**


Figure 1. Methodology for extracting phenophases in GSI module.
Figure 2. Spatial distribution of global FLUXNET sites.
Figure 3. Example images of the canopy phenological changes at the US-MOz site.
Figure 4. Comparison of simulated phenophases by using the DLM-GSI and

DLM-GDD models with the observations derived from the green chromatic

coordinate (GCC) data at the US-MOz site.

Figure 5. Absolute bias comparison between simulated phenophases using the

DLM-GSI and DLM-GDD models at the US-MOz site.

Figure 6. Comparison of simulated phenophases using the DLM-GSI and the

DLM-GDD models with the observations derived from the eddy-covariance

measured gross primary production data at all sites.

Figure 7. Absolute bias comparison between simulated phenophases using the

DLM-GSI and the DLM-GDD models at all sites.

Figure 8. A boxplot of absolute biases for phenophases simulated using the DLM-GSI

and DLM-GDD models.

Figure 9. Comparison of simulated gross primary production using the DLM-GSI and

DLM-GDD models with the observations at all sites.

Figure 10. Histogram comparison of the root mean square error (RMSE) and the

index of agreement (IA) for gross primary production simulation using the





DLM-GSI and DLM-GDD models.
Figure 11. Influence of phenological parameters sensitivity on the growing season
index (GSI) varying (US-Wkg, 2007).






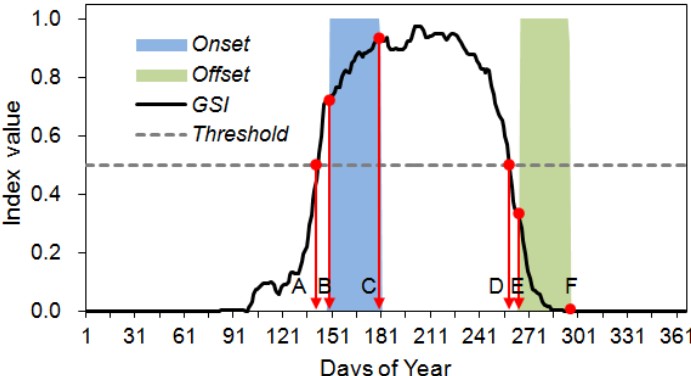


Figure 1. Methodology for extracting phenophases in GSI module. The italics '*Onset*'
and '*Offset*' represent the period of green-up and the period of defoliation,
respectively. The italics '*GSI*' and '*Threshold*' represent the growing season
index and the threshold of GSI, respectively. The letter 'B, C, E, F' represents the
green-up, the normal growth, the defoliation and the dormancy, respectively. The
letter 'A' and 'D' represents the trgger point of green-up and the trgger point of
defoliation, respectively.

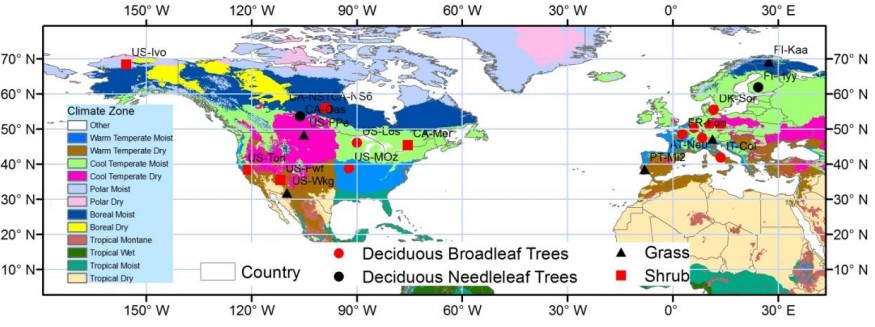


Figure 2. Spatial distribution of global FLUXNET sites.




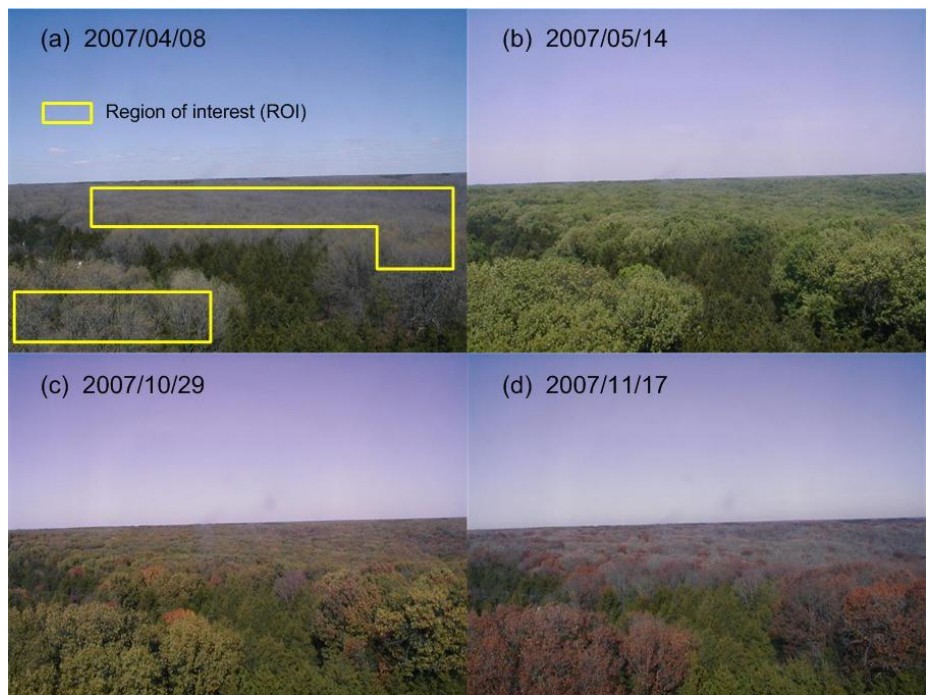


Figure 3. Example images of the canopy phenological changes at the US-MOz site.

The vegetation type in the ROI is the broad-leaf deciduous forest. The letter 'a-d'

represents the green-up, the normal growth, the defoliation and the dormancy,

respectively.






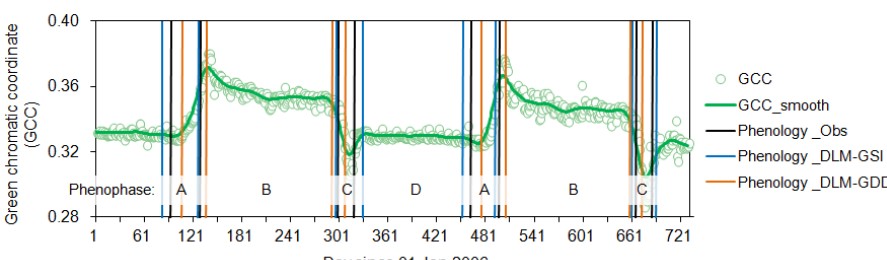


Figure 4. Comparison of simulated phenophases by using the DLM-GSI and

DLM-GDD models with the observations derived from the green chromatic

coordinate (GCC) data at the US-MOz site.


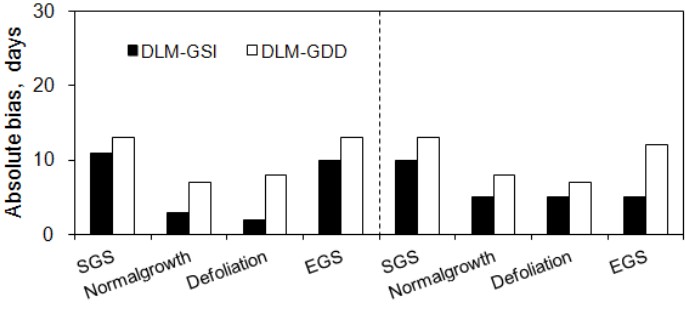


Figure 5. Absolute bias comparison between simulated phenophases using the

DLM-GSI and DLM-GDD models at the US-MOz site. The abbr. 'SGS'

represents the start of growing season, and the 'EGS' means the end of growing

season.






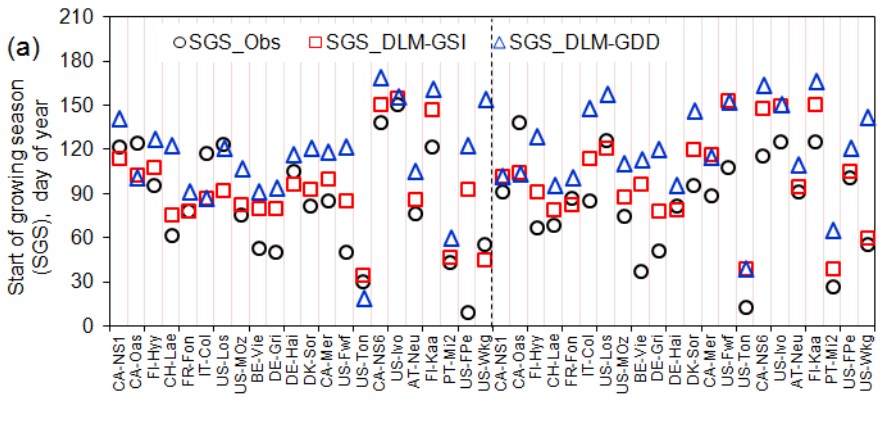



Figure 6. Comparison of simulated phenophases using the DLM-GSI and the

DLM-GDD models with the observations derived from the eddy-covariance

measured gross primary production data at all sites. The letters 'a' and 'b'

represent the start of growing season (SGS) and the end of growing season

(EGS), respectively. All sites in each subfigure contain two consecutive years.







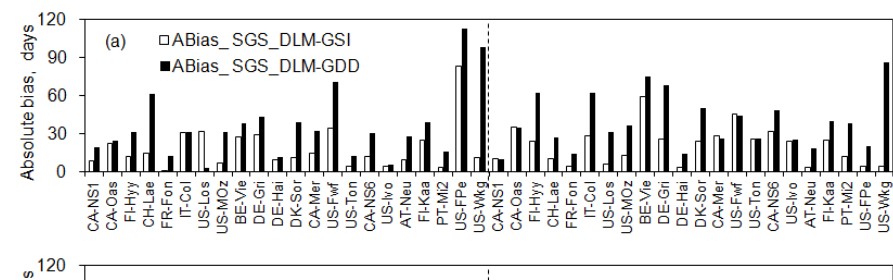


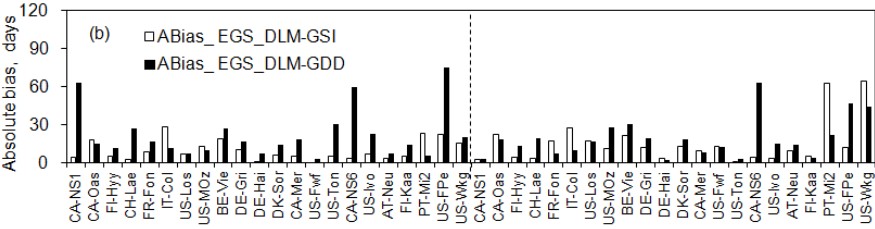

Figure 7. Absolute bias comparison between simulated phenophases using the

DLM-GSI and the DLM-GDD models at all sites. The letters 'a' and 'b'

represent the start of growing season (SGS) and the end of growing season

(EGS), respectively. All sites in each subfigure contain two consecutive years.







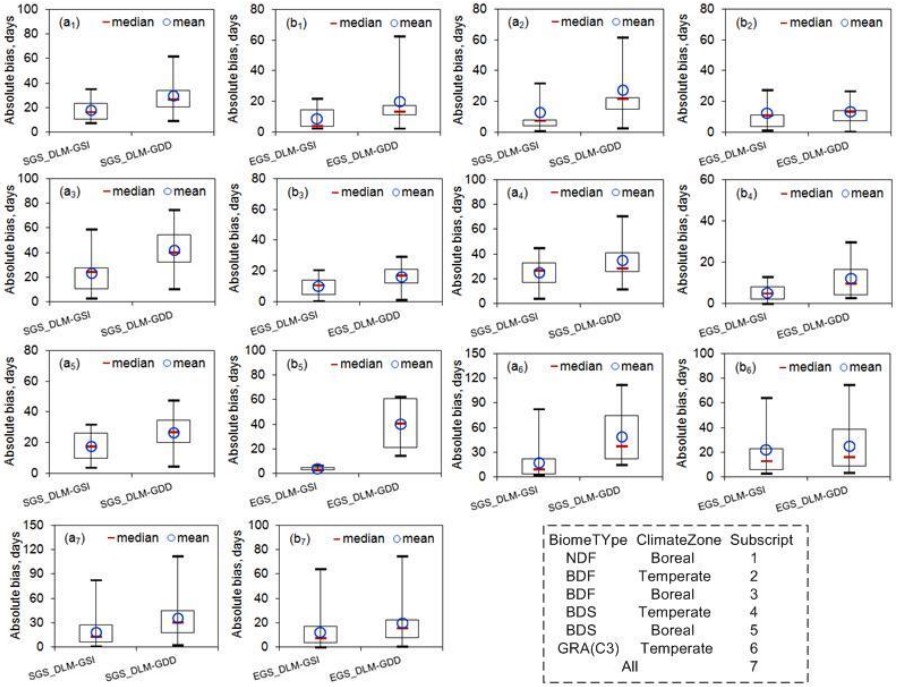

Figure 8. A boxplot of absolute biases for phenophases simulated using the DLM-GSI

and DLM-GDD models. The letters 'a' and 'b' represent the start of growing

season (SGS) and the end of growing season (EGS), respectively. The

abbreviations in the biome types: 'NDF' represents needleleaf deciduous forest;

'BDF' represents broadleaf deciduous forest; 'BDS' represents broadleaf

deciduous shrub; 'GRA' represents grassland.






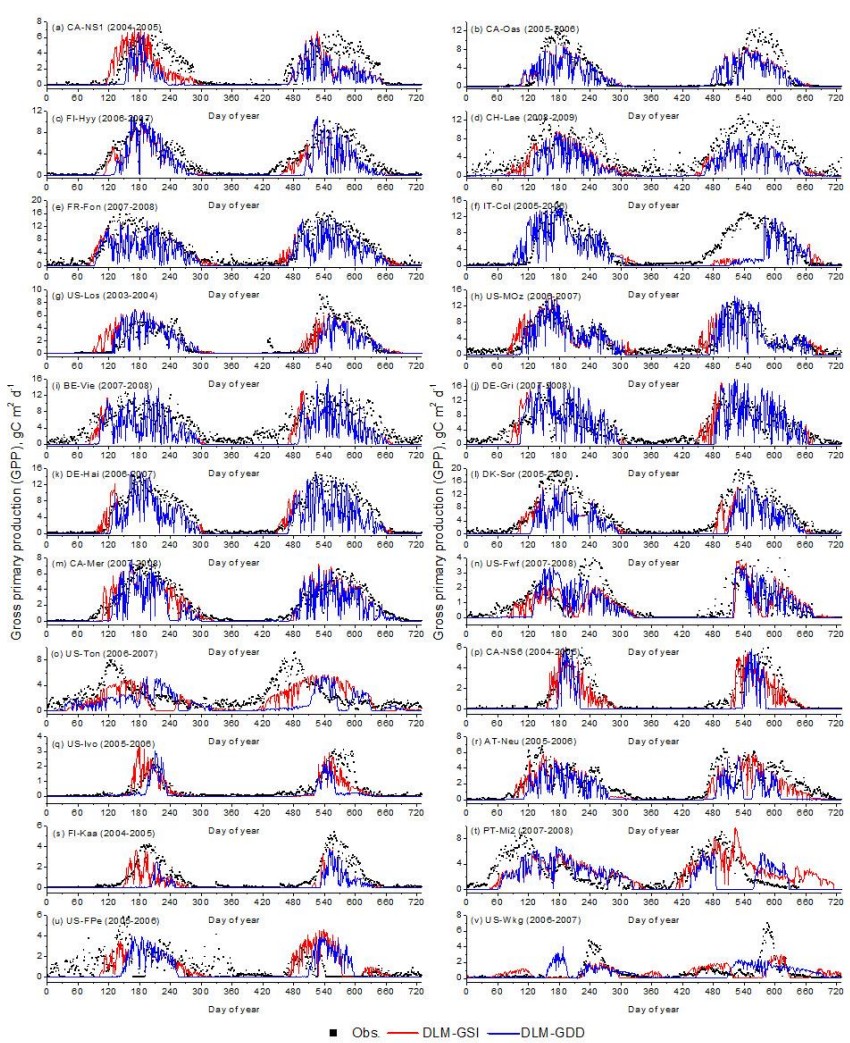


Figure 9. Comparison of simulated gross primary production using the DLM-GSI and

DLM-GDD models with the observations at all sites.




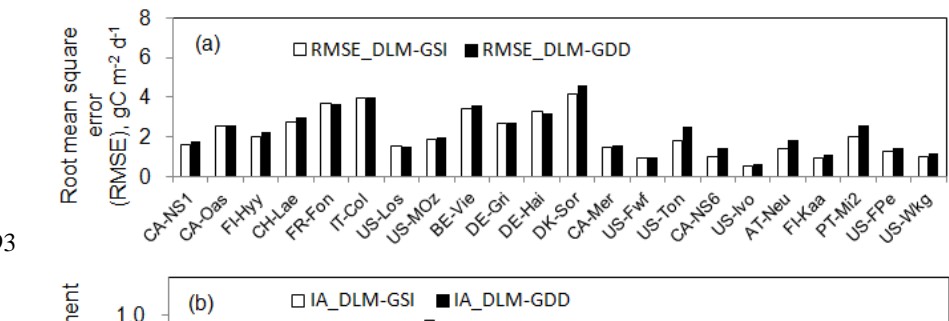


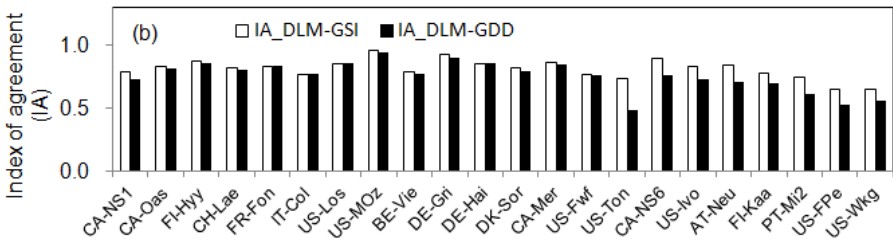


795 Figure 10. Histogram comparison of the root mean square error (RMSE) and the

796   index of agreement (IA) for gross primary production simulation using the

797   DLM-GSI and DLM-GDD models. The letters 'a' and 'b' represent the RMSE

798   and IA, respectively.



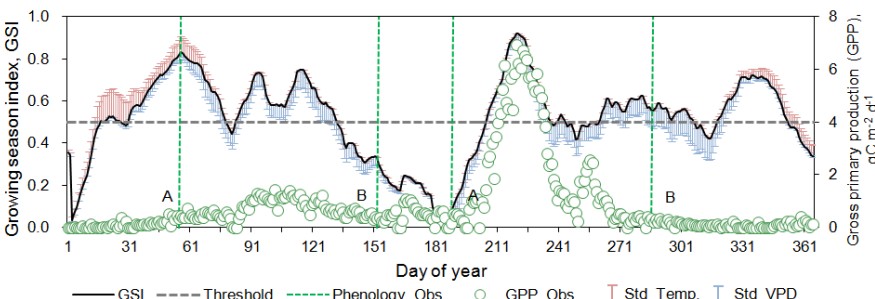

Figure 11. Influence of phenological parameters sensitivity on the growing season

index (GSI) varying (US-Wkg, 2007). The error bars in light red being marked as

positive errors were the sensitivity standard deviation of the temperature. The

error bars in light blue being marked as negative errors were the sensitivity

standard deviation of the vapor pressure deficit (VPD). The letters A and B

represent the start of growing season and the end of growing season, respectively.

The observed phenophases data were derived from the eddy-covariance

measured gross primary production (GPP).



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
