# Peer review of "Improving vegetation phenological parameterization of a land surface model"

_Biogeosciences, 2016_

## Referee Comment (RC1) · Anonymous Referee #1 · 27 May 2016

Chen and Che compared two types of modeling approaches of vegetation leaf phenology: based on Growing Season Index (GSI) or Growing Degree Day (GDD) in the framework of the Dynamic Land Model (DLM). Using GPP data from FLUXNET sites and near-surface remote sensing data from the PhenoCam network as the benchmark, the authors found that DLM-GSI has generally better performance than DLM-GDD and therefore concluded that using GSI phenology model improved DLM. The study itself doesn't have evident flaws, however there is a large room to improve the presentation quality.

Major points: This manuscript may give the readers an impression that GSI model is better than GDD model. But obviously this is not the truth. The authors optimized the GSI model but remain GDD model as default in CLM. I would expect the performance of the GDD model will be similar as the GSI model after optimization. I suggest the

authors should add a paragraph of discussion on this point, with explicit statement that the study doesn't suggest GSI model is better than GDD model. Otherwise it is too ambitious since GDD is a big family of models.

The source of FLUXNET data is missing in this paper. The authors must clarify it. However, FLUXNET has already released the 2015 version data (freely available at http://fluxnet.fluxdata.org/data/fluxnet2015-dataset/), which includes a lot more site years, especially recent-year data. If the authors can use the 2015 data and obviously there would be more GCC data from PhenoCam sites can be involved. It seems to me a weak point that only one PhenoCam site was used in this study.

There are many grammar mistakes through the manuscript. I strongly suggest the authors seek help to polish the written English in this paper.

Specific points: L17-19: but the authors state that GSI model hasn't been used in LSMs in the main text L52: simulating –> simulate L53: delete "change" L59-60: the statement is not right. LAI in CLM can be either prescribed or prognostic. L62: Can you give examples of implicit and explicit+implicit phenology models? L66: starts->originates; add reference to "Reaumur's approach" L79: insert "as" before "important factors" L96-99: please rephrase this sentence. It is not clear whether combining EASS and CLM4 happened firstly or coupling phenology model to DLM happened earlier? L107: common used –> widely-used L112: the authors should provide clear reasons why they considered GPP into the analysis in this paper L118: absorbed –> borrowed L167: Does CTEM use GSI? If yes, why the authors argue that no LSM uses GSI? L168: Maybe I am wrong, but how net photosynthesis can be positive before leafout? L272: requirements –> criteria L275: describe what are Level 3 and Level 4 data? L298: measured –> derived L327: position –> location L328: I understand there is no overlap between PhenoCam data and fluxnet data at most sites, but please explicitly clarify this point to the readers L334-346 should move this paragraph into discussions L354: change the sentence to: the effects of phenology on GPP can be evaluated by using the two model versions.

Discussions: Perhaps the authors can make some comments on possible reasons of that DLM-GSI is better than DLM-GDD in their study. Although it is not a must, I believe it will make the paper more interesting.

---

## Referee Comment (RC2) · Anonymous Referee #2 · 27 May 2016

Chen et al. "Improving vegetation phenological parameterization of a land surface model"

Recommendation: reject

GENERAL

This study evaluates two phenological schemes implemented into a land surface model (LSM). The comparisons show that simulations with GSI scheme perform better than those with GDD scheme. The authors further estimate that simulated GPP with the former scheme shows smaller biases than the latter. Evaluation of phenological schemes for LSM is important. However, this specific study does not commit such purpose due to the lack of scientific contributions and flaws in the analyses.

First, missing of the scientific merits. The main purpose of the study is to compare two phenological schemes. Such inter-comparison has been widely performed (Chuine et al., 1999; Morin et al., 2009; Migliavacca et al., 2012). This study does not add anything new to the scientific community.

Second, flaws in the analyses. There are at least three flaws.

(1) Biased selections of phenological parameters. The authors explained that parameters were adopted from the literature. However, those parameters are appropriate for specific models and/or tree species, but may not be fit for the current study. Without reasonable calibrations, we do not know whether improper parameters contribute to the biases in the schemes. For example, the parameters of GSI scheme were further optimized based on GPP data (Lines 383-384) but those for GDD scheme were adopted directly from CLM model (Lines 388-389). This may explain why the former has better performance.

(2) Improper usage of meteorological forcings. As described in the text (Line 210), the GDD scheme relies on soil temperature while the GSI scheme does not. However, soil variables are adopted from a model instead of observations (Lines 367-369). This also contributes to the biases in GDD approach.

(3) Unnecessary repetition in the comparisons. The authors investigated the impacts of phenological biases on carbon uptake. They found that, relative to GDD scheme, GSI scheme has smaller biases in both phenology and GPP. However, the observations of phenology are derived based on GPP (Line 302). As a result, it is not a surprise that one scheme with better performance for GPP-derived phenology (Figure 6) also has better performance for GPP simulations.

Third, the writing of the paper needs further improvements. I found many redundancies in the text. For example, the last paragraph of section 3 (Lines 557-563) can be replaced with the last sentence, because the whole paragraph is repeating the same conclusion. In addition, most of discussion section lists only problems and uncertainties for current schemes, with limited explorations of the causes, consequences, and/or implications.

SPECIFIC

Line 44-45: Sequence of references should be chronological. Similar problems exist for other citations.

Line 62: How to define "explicit", "implicit", and "both"?

Line 75: Format of the citation should be "(Arora and Boer, 2005)". Similar problems should be corrected.

Line 103: "researched" should be "researches".

Lines 183-188: This paragraph is almost identical to that for spring phenology (Lines 165-170).

Lines 334-346: Most of this paragraph is more appropriate for the discussion section.

Reference

Chuine, I., Cour, P., and Rousseau, D. D.: Selecting models to predict the timing of flowering of temperate trees: implications for tree phenology modelling, Plant Cell Environ, 22, 1-13, doi:10.1046/J.1365-3040.1999.00395.X, 1999.

Migliavacca, M., Sonnentag, O., Keenan, T. F., Cescatti, A., O'Keefe, J., and Richardson, A. D.: On the uncertainty of phenological responses to climate change, and implications for a terrestrial biosphere model, Biogeosciences, 9, 2063-2083, doi:10.5194/Bg-9-2063-2012, 2012.

Morin, X., Lechowicz, M. J., Augspurger, C., O' Keefe, J., Viner, D., and Chuine, I.: Leaf phenology in 22 North American tree species during the 21st century, Global Change Biol, 15, 961-975, doi:10.1111/J.1365-2486.2008.01735.X, 2009.

---

## Author Comment (AC1) · 21 Aug 2016

Response to Anonymous Referee 1 Comment: Chen and Che compared two types of modeling approaches of vegetation leaf phenology: based on Growing Season Index (GSI) or Growing Degree Day (GDD) in the framework of the Dynamic Land Model (DLM). Using GPP data from FLUXNET sites and near-surface remote sensing data from the PhenoCam network as the benchmark, the authors found that DLM-GSI has generally better performance than DLM-GDD and therefore concluded that using GSI phenology model improved DLM. The study itself doesn't have evident flaws, however there is a large room to improve the presentation quality. Major points: This manuscript may give the readers an impression that GSI model is better than GDD model. But obviously this is not the truth. The authors optimized the GSI model but remain GDD model as default in CLM. I would expect the performance of the GDD model will be

similar as the GSI model after optimization. I suggest the authors should add a paragraph of discussion on this point, with explicit statement that the study doesn't suggest GSI model is better than GDD model. Otherwise it is too ambitious since GDD is a big family of models. Response: Thanks for this comments. It is true that the study doesn't suggest GSI model is better than GDD model. The purpose of comparison of the model efficiency of DLM-GSI with DLM-GDD using the in situ data in this study is to assess that how model accuracy will increase if more representation complexity is involved, instead of just simply comparing the performance of GSI with GDD. This suggestions will followed that a paragraph of discussion on this point will added onto the manuscript when we do the revision. The paragraph is as follows. Most widely used land surface models (LSMs) simulate phenophases using prescribed dates empirically derived functions based on cumulative chilling and forcing units (Yang et al., 2012). Recent studies, however, have demonstrated that current representations of phenology in LSMs are not realistic (Melaas et al., 2016). Compared to the observed values, the Abiases of simulated phenophases using the two versions of DLM were not small, although the Abiases of using DLM-GSI were comparatively less, indicating that the two phenology models still must be further developed in future. In addition, the DLM must also be improved, particularly by obtaining more accurate simulated variables as inputs for the phenology model. Richardson et al. (2012) also found a large bias of the predicted onset of spring phenology using LSMs by comparing with in situ data from eddy covariance sites located in North American deciduous forests (early by more than 2 weeks and, in some cases, by as much as 10 weeks). While the sources of the bias was different for each model, overly simplified model representations and overfitting of model coefficients (or both) are widely known sources of error in phenological models (Linkosalo et al., 2008; Melaas et al., 2013). In addition, model-based phenology representations fail to capture local- to regional-scale variability arising from differences in species composition because there are large interspecific differences in leaf-out timing, even when individuals are exposed to the same conditions (Lechowicz, 1984; Murray et al., 1989; Polgar and Primack, 2011; Melaas et al., 2015). Models

range in scope from specific to quite broad. The GDD model simply assumes that the ecosystem phenology is barely controlled by environmental conditions (temperature, moisture and etc.) and more suitable for larger scales, while the GSI tries to additionally consider the factors of ecosystem processes besides the environmental conditions and consequently it is more suitable for smaller scales and needs more information as input. Depending on the complexity of the model, different factors are included or omitted. The purpose of comparison of the model efficiency of DLM-GSI with DLM-GDD using the in situ data in this study is to assess that how model accuracy will increase if more representation complexity is involved, instead of just simply comparing the performance of GSI with GDD. To assess their applicability with certain accuracy, the parameters of the GSI scheme were further optimized using GPP data but those for GDD scheme were simply adopted from the CLM model. As more data on phenological response to climate change emerge, and a better understanding of physiological mechanisms controlling leaf-out develops, more accurate representations of ecosystem dynamics will be possible (Clark et al., 2001; Lebourgeois et al., 2010). Refs related: Yang X, Mustard JF, Tang JW, Xu H (2012) Regional-scale phenology modeling based on meteorological records and remote sensing observations. Journal of Geophysical Research-Biogeosciences, 117.G3, 1–18. Richardson AD, Anderson RS, Arain MA et al. (2012) Terrestrial biosphere models need better representation of vegetation phenology: results from the North American Carbon Program Site Synthesis. Global Change Biology, 18, 566–584. Linkosalo T, Lappalainen HK, Hari P (2008) A comparison of phenological models of leaf bud burst and flowering of boreal trees using independent observations. Tree Physiology, 28, 1873–1882. Melaas EK, Friedl MA, Zhu Z (2013) Detecting interannual variation in deciduous broadleaf forest phenology using Landsat TM/ETM plus data. Remote Sensing of Environment, 132, 176–185. Melaas EK, Friedl MA, Richardson AD (2016) Multiscale modeling of spring phenology across Deciduous Forests in the Eastern United States, Global Change Biology, 22, 792–805.

Lechowicz MJ (1984) Why do temperate deciduous trees leaf out at different times –

adaptations and ecology of forest communities. American Naturalist, 124, 821–842. Murray MB, Cannell MGR, Smith RI (1989) Date of budburst of 15 tree species in Britain following climatic warming. Journal of Applied Ecology, 26,693–700. Polgar and Primack (2011) Leaf-out phenology of temperate woody plants: from trees to ecosystems, New Phytologist, 191, 926–941. Lebourgeois F, Pierrat JC, Perez V, Piedallu C, Cecchini S, Ulrich E (2010) Simulating phenological shifts in French temperate forests under two climatic change scenarios and four driving global circulation models. International Journal of Biometeorology 54, 563–581. Clark JS, Carpenter SR, Barber M, Collins S, Dobson A, Foley JA, Lodge DM, Pascual M, Pielke R, Pizer W et al. (2001) Ecological forecasts: an emerging imperative. Science 293, 657–660.

Comments: The source of FLUXNET data is missing in this paper. The authors must clarify it. However, FLUXNET has already released the 2015 version data (freely available at http://fluxnet.fluxdata.org/data/fluxnet2015-dataset/), which includes a lot more site years, especially recent-year data. If the authors can use the 2015 data and obviously there would be more GCC data from PhenoCam sites can be involved. It seems to me a weak point that only one PhenoCam site was used in this study. Response: The source of FLUXNET data was given as (http://fluxnet.ornl.gov/) (see Line 271 Page 14 please). When we prepare this paper the fluxnet data for 2015 were not available. We will further use these valuable data in our future work.

Comments: There are many grammar mistakes through the manuscript. I strongly suggest the authors seek help to polish the written English in this paper. Specific points: L17-19: but the authors state that GSI model hasn't been used in LSMs in the main text L52: simulating –> simulate L53: delete "change" L59-60: the statement is not right. LAI in CLM can be either prescribed or prognostic. L62: Can you give examples of implicit and explicit+implicit phenology models? L66: starts- >originates; add reference to "Reaumur's approach" L79: insert "as" before "important factors" L96-99: please rephrase this sentence. It is not clear whether combining EASS and CLM4 happened firstly or coupling phenology model to DLM happened earlier? L107: common used

–> widely-used L112: the authors should provide clear reasons why they considered GPP into the analysis in this paper L118: absorbed –> borrowed L167: Does CTEM use GSI? If yes, why the authors argue that no LSM uses GSI? L168: Maybe I am wrong, but how net photosynthesis can be positive before leafout? L272: requirements –> criteria L275: describe what are Level 3 and Level 4 data? L298: measured –> derived L327: position –> location L328: I understand there is no overlap between PhenoCam data and fluxnet data at most sites, but please explicitly clarify this point to the readers L334-346 should move this paragraph into discussions L354: change the sentence to: the effects of phenology on GPP can be evaluated by using the two model versions. Response: Many thanks. All these suggestions will be followed when we do the revision.

Comments: Discussions: Perhaps the authors can make some comments on possible reasons of that DLM-GSI is better than DLM-GDD in their study. Although it is not a must, I believe it will make the paper more interesting. Response: Yes, this suggestion will be followed. Thanks. See our response to your "Major points".

Please also note the supplement to this comment:
http://www.biogeosciences-discuss.net/bg-2016-165/bg-2016-165-AC1-
supplement.pdf

---

## Author Comment (AC2) · 22 Aug 2016

Response to Anonymous Referee 2 Anonymous Referee 2 GENERAL This study evaluates two phenological schemes implemented into a land surface model (LSM). The comparisons show that simulations with GSI scheme perform better than those with GDD scheme. The authors further estimate that simulated GPP with the former scheme shows smaller biases than the latter. Evaluation of phenological schemes for LSM is important. However, this specific study does not commit such purpose due to the lack of scientific contributions and flaws in the analyses. First, missing of the scientific merits. The main purpose of the study is to compare two phenological schemes. Such inter-comparison has been widely performed (Chuine et al., 1999; Morin et al., 2009; Migliavacca et al., 2012). This study does not add anything new to the scientific community. Response: We must clarify that the main purpose of the study is not

to compare the two phonological schemes. The purpose of comparison of the model efficiency of DLM-GSI with DLM-GDD using the in situ data in this study is to assess that how model accuracy will increase if more representation complexity is involved, instead of just simply comparing the performance of GSI with GDD. Morin et al. (2009) developed models predicting the date of leaf unfolding (date of first fully formed leaf) for 18 North American temperate tree species representing 11 genera. In the paper of Chuine et al. (1999), The Spring–Warming model (Cannell Smith, 1983), SeqSar and Par1Sar models (Chuine et al., 1999) were used. All of these three models are functions of temperature or based on the sum of degree-days. The models used in the paper of Migliavacca et al. (2012), are two main categories of models. While the model categories differ in their assumptions of how warm and cold temperatures control developmental processes, none of them belongs to the kind of the GSI model. That is to say, none of the three reference papers performed the phenology schemes comparison. Lack of inter-comparison of phenology schemes can be found in the literature. We cannot agree with the referee that "such inter-comparison has been widely performed".

Comment: Second, flaws in the analyses. There are at least three flaws. (1) Biased selections of phenological parameters. The authors explained that parameters were adopted from the literature. However, those parameters are appropriate for specific models and/or tree species, but may not be fit for the current study. Without reasonable calibrations, we do not know whether improper parameters contribute to the biases in the schemes. For example, the parameters of GSI scheme were further optimized based on GPP data (Lines 383-384) but those for GDD scheme were adopted directly from CLM model (Lines 388-389). This may explain why the former has better performance. (2) Improper usage of meteorological forcings. As described in the text (Line 210), the GDD scheme relies on soil temperature while the GSI scheme does not. However, soil variables are adopted from a model instead of observations (Lines 367-369). This also contributes to the biases in GDD approach. (3) Unnecessary repetition in the comparisons. The authors investigated the impacts of phenological biases on carbon uptake. They found that, relative to GDD scheme, GSI scheme has smaller

biases in both phenology and GPP. However, the observations of phenology are derived based on GPP (Line 302). As a result, it is not a surprise that one scheme with better performance for GPP-derived phenology (Figure 6) also has better performance for GPP simulations. Response: I will respond to the above three points, respectively, as below: (1) The parameters of the GSI scheme were further optimized using GPP data but those for GDD scheme were simply adopted from the CLM model. Why we did so is to assess the applicability of these two phenology schemes with certain accuracy. The GDD model was simply parameterized in order to test how well it will perform when it is applied to large scales with less proper input information available. The objective of coupling DLM with GSI is to develop a more specific and local scale model, so the parameters of the DSI model were initialized and optimized based on in situ measurements. (2) The measured soil temperature is not always available. And also for testing the capability of GDD, we used the modelled soil temperature as model inputs. Actually, the phenology model is coupled with land surface model. Few studies used measured soil temperature data as phenology model inputs. ïijĹ3ïijĽWe partly agree that the phenology model initialization using GPP data would improve GPP simulation accuracy. It is not necessary that the GPP simulation with DLM-GSI must better than DLM-GDD though the former used measured GPP to initialize its parameters. In this study we aim at quantifying the improvement in GPP modeling using calibrated GSI comparing with those using the default parameters of GDD. We need this model comparison research to develop/improve a coupled dynamic land surface model suitable for a scope of local to intermedia scales and for different research purposes.

Comment: Third, the writing of the paper needs further improvements. I found many redundancies in the text. For example, the last paragraph of section 3 (Lines 557-563) can be replaced with the last sentence, because the whole paragraph is repeating the same conclusion. In addition, most of discussion section lists only problems and uncertain ties for current schemes, with limited explorations of the causes, consequences, and/or implications. SPECIFIC Line 44-45: Sequence of references should be chronological. Similar problems exist for other citations. Line

62: How to define "explicit", "implicit", and "both"? Line 75: Format of the citation should be "(Arora and Boer, 2005)". Similar problems should be corrected. Line 103: "researched" should be "researches". Lines 183-188: This paragraph is almost identical to that for spring phenology (Lines 165-170). Lines 334-346: Most of this paragraph is more appropriate for the discussion section. Reference Chuine, I., Cour, P., and Rousseau, D. D.: Selecting models to predict the timing of flowering of temperate trees: implications for tree phenology modelling, Plant Cell Environ, 22, 1-13, doi:10.1046/J.1365-3040.1999.00395.X, 1999. Migliavacca, M., Sonnentag, O., Keenan, T. F., Cescatti, A., O'Keefe, J., and Richardson, A. D.: On the uncertainty of phenological responses to climate change, and implications for a terrestrial biosphere model, Biogeosciences, 9, 2063-2083, doi:10.5194/Bg-9-2063-2012, 2012. Morin, X., Lechowicz, M. J., Augspurger, C., O' Keefe, J., Viner, D., and Chuine, I.: Leaf phenology in 22 North American tree species during the 21st century, Global Change Biol, 15, 961-975, doi:10.1111/J.1365-2486.2008.01735.X, 2009. Response: Many thanks. We will carefully improve the writing of the paper when we revise the paper. SPECIFIC Line 44-45: Sequence of references should be chronological. Similar problems exist for other citations. This suggestion will be followed in the revised version. Line 62: How to define "explicit", "implicit", and "both"? Thanks. It will corrected as "...which is embedded in LSMs either explicitly or implicitly. Line 75: Format of the citation should be "(Arora and Boer, 2005)". Similar problems should be corrected. Line 103: "researched" should be "researches". Thanks. The suggestion will follow. Lines 183-188: This paragraph is almost identical to that for spring phenology (Lines 165-170). These two paragraphs discuss "leaf green-up" and "leaf defoliation", respectively. I will rewrite this paragraph in the revised version. Lines 334-346: Most of this paragraph is more appropriate for the discussion section. Thanks for this comment. We will follow your suggestion to move this paragraph to discussion section.

Please also note the supplement to this comment:
http://www.biogeosciences-discuss.net/bg-2016-165/bg-2016-165-AC2-

supplement.pdf